

# How "dirty" is the Dark Triad? Dark character profiles, swearing, and sociosexuality

Danilo Garcia

[1] Department of Behavioral Sciences and Learning, Linköping University, Linköping, Sweden
[2] Centre for Ethics, Law and Mental Health (CELAM), Institute of Neuroscience and Physiology, University of Gothenburg, Sweden
[3] Department of Psychology, University of Gothenburg, Gothenburg, Sweden
[4] Blekinge Center of Competence, Region Blekinge, Karlskrona, Sweden
[5] Network for Well-Being, Sweden

Corresponding author
Danilo Garcia,
danilo.garcia@icloud.com

## ABSTRACT

Malevolent character traits (i.e., the Dark Triad: Machiavellianism, narcissism, and psychopathy) are associated to emotional frigidity, antagonism, immoral strategic thinking, betrayal, exploitation, and sexual promiscuity. Despite the fact that character is a complex adaptive system, almost every study has solely investigated the linear association between malevolent character and attitudes towards both swearing and sociosexual orientation (i.e., behavior, attitude, and desire regarding promiscuous sexual behavior). In contrast, the aim in this set of studies was to evaluate these associations in relation to specific profiles of malevolent character (i.e., the Dark Cube). In two studies participants responded to the Dark Triad Dirty Dozen, the Taboo Words' Offensiveness and Usage Inventories (i.e., attitudes towards 30 swear words' level of offensiveness and usage) (Study 1: $N_1 = 1,000$) and the Sociosexual Orientation Inventory Revised (Study 2: $N_2 = 309$). Participants were clustered according to all eight possible combinations based on their dark trait scores (M/m = high/low Machiavellianism; N/n = high/low narcissism; P/p = high/low psychopathy). The results of this nonlinear approach suggested that the frequent usage, not level of offensiveness, of swear words was associated to Machiavellianism and narcissism. In other words, individuals with high levels in these traits might swear and are verbally offensive often, because they do not see swearing as offensive (cf. with the attitude-behavior-cognition-hypothesis of taboo words; Rosenberg, Sikström & Garcia, 2017). Moreover, promiscuous sociosexual attitude and desire were related to each dark trait *only* when the other two were low. Additionally, promiscuous sociosexual behavior was not associated to these malevolent character traits. That is, individuals high in the dark traits are willing to and have the desire to engage in sexual relations without closeness, commitment, and other indicators of emotional bonding. However, they do not report high levels of previous sexual experience, relationships, and infidelity. Hence, they approve and desire for it, but they are not actually doing it. The use of person-centered and non-linear methods, such as the Dark Character Cube, seem helpful in the advancement of a coherent theory of a biopsychosocial model of dark character.

# INTRODUCTION

*"I love French wine, like I love the French language. I have sampled every language, French is my favorite. Fantastic language. Especially to curse with: Nom de dieu de putain de bordel de merde de saloperie de connard d'enculé de ta mère.*

*It's like wiping your arse with silk. I love it".*

*"Choice is an illusion created between those with power and those without".*

*"Please, ma cherie. I have told you. We are all victims of causality. I drank too much wine, I must take a piss. Cause and effect".*

*—The Merovingian in Matrix Reloaded (Silver, Wachowski & Wachowski, 2003)*

Dark personality traits are expressed as manipulativeness, a cynical worldview and lack of morality (i.e., Machiavellianism), a sense of grandiosity and vulnerable self-esteem (i.e., narcissism), and also low conscientiousness, high impulsivity, and high levels of thrill-seeking behavior (i.e., psychopathy). Although each of these behaviors are associated with a specific dark character trait (*Paulhus & Williams, 2002*; *Furnham, Richards & Paulhus, 2013*; *Hare, 1985*; *Jones & Paulhus, 2009*; *Jones & Paulhus, 2014*), individuals high in any of the dark character traits tend to be uncooperative and unagreeable (*Garcia et al., 2015*; *Garcia & Rosenberg, 2016*; *Kajonius et al., 2016*). In other words, malevolent character seems to be a form of immature character[1] expressed as being unempathetic, lacking self-control, and having low "moral intuition". Individuals high in these dark traits often use violence (both verbal and physical) and also promiscuous and "dirty" behavior to manipulate or submit others in order to gain power or fulfill own desires (i.e., an outlook of separation), hence, they lack a sense for cooperation and altruism (i.e., an outlook of unity; *Cloninger, 2004*).

For instance, swear words, which main purpose is to express emotions, especially anger and frustration (*Jay, 2000*; *Jay & Janschewitz, 2008*), are more frequently used by individuals who are higher in the dark character traits. More specifically, words that express anger and negative emotion seem to be more frequently used by individuals high in Machiavellianism and psychopathy, while words related to sex are more frequently used by individuals high in narcissism (*Sumner et al., 2012*). Importantly, a wide range of research suggest that swearing, in moderation, may increase pain tolerance (*Stephens, Atkins & Kingston, 2009*) and even reduce stress (*Byrne, 2018*). Hence, it is important to understand the mechanism behind swearing in relation to dark personality traits. In this context, according to the (A)attitude-(B)behavior-(C)cognition-hypothesis of taboo words, the level of offensiveness of swear words predicts how often people swear (*Rosenberg, Sikström & Garcia, 2017*). Thus, the way the individual perceives how offensive the word is and how often she/he uses the swear word are two separate features that might relate differently to high levels of the dark traits. If it is so, individuals who score high in the dark traits are expected to swear often and to not see the swearing words they use as offensive.

Furthermore, other type of "dirty" behavior related to malevolent traits can be observed with regard to a person's attitude towards sexual life. Individuals, specially males, who are high in the dark traits use an exploitative short-term mating strategy,[2] that is, an strategy

[1] See for example *Cloninger (2004)*, who defines a mature character as high levels in three character traits: self-directedness, cooperativeness, and self-transcendence (see also *Garcia & Rosenberg, 2016*).

[2] According to evolutionary psychology, human mating strategies tend to range from short-term relationships to long-term relationships characterized by little and heavy commitment, respectively (*Buss, 2019*).

with tactics to avoid entangling commitments (*Jonason et al., 2009*; *Jonason, Li & Czarna, 2013*; *Jonason & Buss, 2012*; *González Moraga, Nima & Garcia, 2017*). In this context, sociosexuality or differences in a person's willingness to engage in sexual relations without closeness, commitment, and other indicators of emotional bonding, can be understood in three different components: sociosexual behavior (i.e., individuals' previous sexual experience, relationships, and infidelity), sociosexual attitude (i.e., aspects of behavior and desire influenced by moral feelings, reflections and self-presentation based on values, habits and social effects), and sociosexual desire (i.e., the notion of a dispositional motivation that refers to effort given to temporary and long-term sexual relationships) (*Penke & Asendorf, 2008*; *Simpson & Gangestad, 1992*). Importantly, past findings on the main effects of high levels of socosexuality on people's health range from negative to positive to even nonsignificant (*Vrangalova & Ong, 2014*).

In sum, the relationship between "dirty" behavior and malevolent character is not consistent. One probable part of this shortcoming might be the current understanding of traits as the basic unit of personality (*Cloninger & Zwir, 2018*; *Cloninger et al., 2020*). Most of the current studies on dark traits, if not all, conduct some type of association analyses between the malevolent dark traits and different outcomes. Molecular studies, however, indicate that the basic unit of personality is actually profiles, not traits (see *Zwir et al., 2018a*; *Zwir et al., 2018b*; *Zwir et al., 2019a*; *Zwir et al., 2019b*; *Cloninger & Cloninger, 2019*). Therefore, the purpose of the present study is to test a relatively new approach, the Dark Cube, to study the relationship between the dark traits and peoples' swearing and sociosexuality. The question is if this approach adds any new information to the common linear correlation approach.

## THE DARK CUBE

The Dark Cube (*Garcia & Rosenberg, 2016*; *Garcia, 2018*; *Garcia & González Moraga, 2017*; *Garcia et al., 2018*) is based on the presupposition that the Dark Triad is composed of overlapping yet distinctive constructs that can be measured separately (*Paulhus & Williams, 2002*) but that may vary within the individual. Hence, the Dark Cube consists of all eight possible combinations of high/low scores in the three dark traits (Fig. 1). This is one way of addressing personality traits as a whole system unit or a dynamic complex adaptive system (cf. *Cloninger, 2004*). This approach allows for the nonlinear investigation of, in this case, malevolent character profiles (see also *Bergman & Wångby, 2014*; *Bergman & Magnusson, 1997*; *Cloninger, Svrakic & Svrakic, 1997*). For instance, despite the fact that some studies using the Big Five traits (*Costa Jr, McCrae & Dye, 1991*; *John & Srivastava, 1999*; *González Moraga, 2015*) have found associations between, for example, neuroticism and some of the dark traits, these associations are not consistent in the literature (*Vernon et al., 2008*). Studies using the Dark Cube on the other hand, show that Big Five traits and the malevolent characters traits are associated *only* under certain conditions, thus, probably explaining the inconsistencies in the literature (*Garcia, 2018*). For example, while high levels of narcissism are associated to high levels of extraversion and high levels of psychopathy are associated to high levels of low agreeableness per se; high Machiavellianism was associated to high

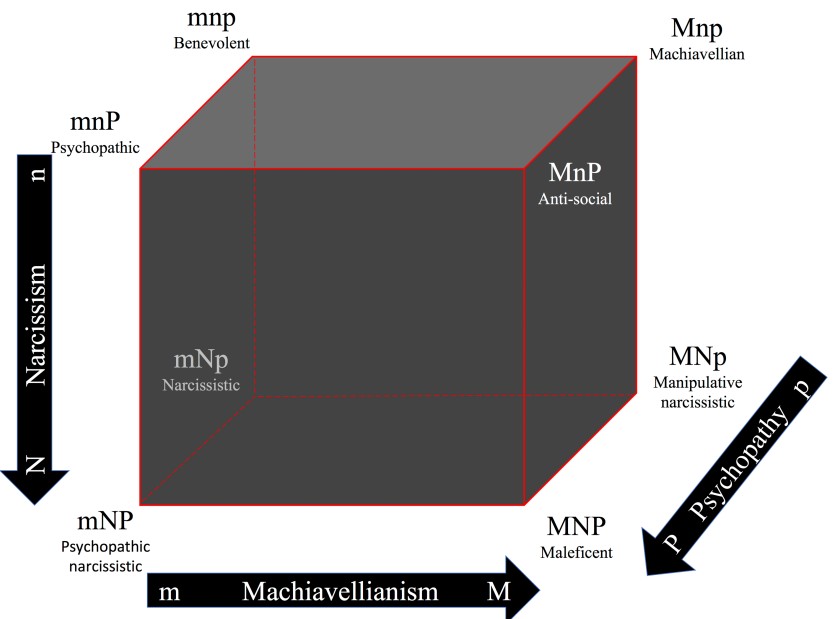

**Figure 1** **The Dark Cube as an analogy to Cloninger's character cube, showing all eight possible combinations of high/low scores in Machiavellianism, narcissism, and psychopathy.** Note: adapted with permission from CR Cloninger. The directions of the arrows represent higher values. M/m, high/low Machiavellianism; N/n, high/low narcissism; P/p, high/low psychopathy. Originally published in: *Garcia & Rosenberg (2016)*.

neuroticism *only* when both narcissism and psychopathy were low, high narcissism and high conscientiousness were associated *only* when both Machiavellianism and psychopathy were also high, and high psychopathy and high neuroticism were associated *only* when Machiavellianism was low and narcissism was high (*Garcia & González Moraga, 2017*). In other words, the Dark Cube analyses showed how the complex interactions between dark traits within the individual, an interaction that is often ignored in a majority of studies, may explain inconsistencies, such as, why high levels of psychopathy are in some studies associated to high levels of neuroticism and not associated at all in others (*Garcia et al., 2015*). More specifically, the Dark Cube analyses suggest that individuals high in psychopathy have a proneness to worry, rumination, hostility, sadness, hopelessness, impulsiveness, and sensitivity in social situations (i.e., high neuroticism) *if* they are also low in manipulative tendencies and highly narcissistic at the same time (*Garcia & González Moraga, 2017*). Hence, the Dark Cube, with its eight dark malevolent profiles, is a tool that might help to clarify some of the mixed and inconsistent associations in the Dark Triad literature (*Garcia, 2018*).

## The present set of studies

The aim in this set of studies was to evaluate the interactions among specific combinations of malevolent character traits in relation to swearing (Study 1) and sociosexuality (Study 2). The question is *if* the Dark Cube approach adds more information to the common linear analyses used in most studies. More specifically, this set of studies comprises the

investigation of the relationship between malevolent character traits and "dirty" behavior (i.e., swearing and sociosexual orientation) using non-linear methods (i.e., comparisons between individuals who differed in one malevolent character trait but were alike in the other). This approach considers the complexity of processes within the person (cf. *Cloninger & Zohar, 2011*). The multidimensionality of swearing (how offensive a word is vs. how often the word is used) and sociosexuality (behavior, attitude, and desire) was expected to be a function of a complex interaction: the same malevolent character trait could lead to different "dirty" behavior (i.e., multi-finality), but also that different malevolent character traits could lead to the same "dirty" behavior (i.e., equifinality) (cf. *Cicchetti & Rogosch, 1996*).

## ETHICS STATEMENT

Since the study did not involve a physical intervention and no information on individual health issues was involved in the study, there was no need to involve the ethical board, according to Swedish law at the time the data was collected (2014-2015). The World Medical Association Declaration of Helsinki (1964) was followed. Participants, workers from the crowdsourcing platform Amazon's Mechanical Turk, provided their consent by simply accepting the task (or HIT as it's called in Amazon Mechanical Turk) and then starting to answer the survey. This acceptance is recorded electronically together with the participants' answers.

## STUDY 1: SWEARING

### Method

#### Participants and procedure

Participants in Study 1 were recruited from Amazon's Mechanical Turk and got paid $ 0.50 dollars for taking the survey (data from *Rosenberg, Sikström & Garcia, 2017*). Participants answered to measures of the Dark Triad and both perception and usage of swear words, and also to demographic questions (e.g., gender, age) and two control questions (e.g., "In this question please answer Neither agree or disagree"). A total of 50 participants were removed (i.e., 4.76%) from the final sample due to erroneous answer to one or both control questions. This final sample consisted of 1,000 US-residents, 333 women and 667 men ($N_1$ = 1,000, $M_{age}$ = 31.50 ± 10.27).

#### Measures

*Dark Traits.* The Dark Triad Dirty Dozen (*Jonason & Webster, 2010*) consists of 12 statements (1 = *strongly disagree*, 7 = *strongly agree*), four statements for each dark trait: Machiavellianism ("I tend to manipulate others to get my way"; *Cronbach's α* = .78), narcissism ("I tend to want others to admire me"; *Cronbach's α* = .77), and psychopathy ("I tend to be unconcerned with the morality of my actions"; *Cronbach's α* = .76).

   *Swear Words' Offensiveness.* The Taboo Words' Offensiveness Inventory (*Rosenberg, Sikström & Garcia, 2017*) asks participants to rate (1 = *not offensive at all*, 5 = *very offensive*) how offensive they perceived 30 frequently used swear words (e.g., "fuck",

"shit", "bitch", "cunt", "damn", and "asshole"). An offensiveness score was computed by simply summarizing the average value of the 30 taboo words (*Cronbach's α* = .95).

*Swear Words' Frequency.* The Taboo Words' Usage Inventory (*Rosenberg, Sikström & Garcia, 2017*) asks participants to rate (1 = *rarely or never*, 5 = *very often*) how often they use each one of the 30 frequently used swear words from the Taboo Words' Offensiveness Inventory (e.g., "fuck", "shit", "bitch", "cunt", "damn", and "asshole"). A frequency score was computed by simply summarizing the average value of the 30 taboo words (*Cronbach's α* = .94).

### Statistical procedure

The scores in each dark trait were first transformed to *percentiles* and then used to divide subjects into high and low *percentiles* in each of the three dark traits: Machiavellianism, narcissism, and psychopathy (see *Garcia, 2018*). Then the participants were clustered according to all the possible combinations of high/low scores in Machiavellianism (M/m), narcissism (N/n), and psychopathy (P/p) to define the eight possible Dark Triad profiles: MNP "maleficent" ($n_1$ = 247, 24.7%), MNp "manipulative narcissistic" ($n_1$ = 78, 7.8%), MnP "anti-social" ($n_1$ = 124, 12.4%), Mnp "Machiavellian" ($n_1$ = 52, 5.2%), mNP "psychopathic narcissistic" ($n_1$ = 72, 7.2%), mNp "narcissistic" ($n_1$ = 122, 12.2%), mnP "psychopathic" ($n_1$ = 99, 9.9%), and mnp "benevolent" ($n_1$ = 206, 20.6%).

### Results and discussion

Paired *t-tests* were used to investigate the differences in perception and usage of swear words between individuals with malevolent character profiles who differed in one of the dark character traits but were similar in the other two. In addition, correlation analyses were also conducted in order to investigate the added value of the profile analyses in relation to linear correlations between the dark traits and both swear word's offensiveness and usage.

While the correlation analyses indicated a significant positive correlation between Machiavellianism and swear word usage ($r$ = .31, $p < .001$) and a significant but very low negative correlation between Machiavellianism and swear word offensiveness ($r$ = −.15, $p < .001$), the Dark Cube analyses indicated a more complex relationship (see Table 1). For instance, high levels of Machiavellianism were associated to high levels of swear word usage and to low offensiveness when both narcissism and psychopathy were low (Mnp vs. mnp). Moreover, Machiavellianism was also associated to high swear word usage when narcissism was low and psychopathy was high (MnP vs. mnP) and also when both narcissism and psychopathy were high (MNP vs. mNP). In other words, individuals high in Machiavellianism seem to use swear words very frequently. The *only* exception is when narcissism is high and psychopathy is low (MNp vs. mNp). This specific finding was only discerned when the dark profiles were analyzed and suggests that a sub-group of individuals with a tendency for manipulativeness, a cynical worldview and lack morality (i.e., high Machiavellianism) do not use swear words frequently if they at the same time have a sense of grandiosity and vulnerable self-esteem (i.e., high narcissism) and are high in conscientiousness and low in impulsivity (i.e., low psychopathy). In addition, another specific finding from the Dark Cube analyses, was that individuals with a Machiavellian

**Table 1** Results from the *t*-tests for each Dark Triad character trait for swear words' frequency and offensiveness. The black cells indicate significant results.

| Dark Trait | Dark Profile | Swear Words' Frequency | | | Swear Words' Offensivenes | | |
|---|---|---|---|---|---|---|---|
| | | *t* | *p* | Cohen's *d* | *t* | *p* | Cohen's *d* |
| Machiavellianism | MNP vs. mNP | 2.89 | .004 | 0.32 | −0.60 | .55 | −0.07 |
| | MNp vs. mNp | 0.61 | .54 | 0.09 | 0.54 | .59 | 0.08 |
| | MnP vs. mnP | 2.96 | .003 | 0.40 | −0.75 | .46 | −0.10 |
| | Mnp vs. mnp | 2.64 | .009 | 0.33 | −2.15 | .03 | −0.27 |
| Narcissim | MNP vs. MnP | 0.40 | .69 | 0.04 | −0.23 | .82 | −0.02 |
| | MNp vs. Mnp | −0.40 | .69 | −0.07 | 2.26 | .03 | 0.40 |
| | mNP vs. mnP | 0.51 | .61 | 0.08 | −0.36 | .72 | −0.06 |
| | mNp vs. mnp | 2.13 | .03 | 0.24 | −0.31 | .76 | −0.03 |
| Psychopathy | MNP vs. MNp | 3.79 | .000 | 0.42 | −3.41 | .001 | −0.38 |
| | MnP vs. Mnp | 2.30 | .02 | 0.35 | −0.06 | .95 | −0.01 |
| | mNP vs. mNp | 1.37 | .17 | 0.20 | −1.85 | .07 | −0.27 |
| | mnP vs. mnp | 3.07 | .002 | 0.35 | −1.87 | .06 | −0.21 |

**Notes.**

M/m, high/low Machiavellianism; N/n, high/low narcissism; P/p, high/low; MNP, maleficent; MNp, manipulative narcissistic; MnP, anti-social; Mnp, Machiavellian; mNP, psychopathic narcissistic; mNp, narcissistic; mnP, psychopathic; mnp, benevolent.

profile (Mnp), compared to individuals with a benevolent profile (mnp), actually were the ones that really do not have a problem with swearing, they both use it frequently (Cohen's $d = 0.33$) and found the 30 swear words less offensive (Cohen's $d = −0.27$). The Machiavellian profile was not common in this sample (only 5.2%), but in a sample of 18,192 individuals the Machiavellian profile was almost twice as common (9.60%; *Garcia, 2018*), thus, depending on the sample composition, this might influence the findings when linear analyses are implemented.

With regard to narcissism, the correlation analyses indicated a significant positive but very low correlation between narcissism and swear word usage ($r = .16$, $p < .001$) and no significant correlation between narcissism and swear word offensiveness ($r = −.01$, $p = .802$). The Dark Cube analyses (see Table 1), however, indicated that the level of offensiveness perceived in the swear words was associated to high levels of narcissism *only* when Machiavellianism was high and psychopathy was low (MNp vs. Mnp). This association was positive and moderate (Cohen's $d = 0.40$). That is, in contrast to the correlation analyses, these analyses indicated that individuals with high levels of grandiosity and vulnerable self-esteem (i.e., high narcissism) found the 30 swear words more offensive if they were highly manipulative (high Machiavellianism) and high in conscientiousness and low in impulsivity (i.e., low psychopathy). Nevertheless, high narcissism was associated to high levels of swear word usage when both Machiavellianism and psychopathy were low (mNp vs. mnp). Hence, narcissism seem to have an effect in both high frequency usage of swear words and high offensiveness of swear words depending on the interaction between the other two dark traits within the person.

Finally, with regards to psychopathy, the correlation analyses indicated a significant positive correlation between psychopathy and swear word usage ($r = .28$, $p < .001$) and

a significant but very low negative correlation between psychopathy and swear word offensiveness ($r = -.19$, $p < .001$). The Dark Cube analyses (see Table 1) indicated that high levels of psychopathy were indeed associated to high levels of swear words' usage in most of the cases (i.e., three out of four comparisons: MNP vs. MNp; MnP vs. Mnp; mnP vs. mnp). Nevertheless, these analyses also indicated that when Machiavellianism was low and narcissism was high (mNP vs. mNp), then high psychopathy was not significantly associated to high usage of swear words. That is, individuals who are impulsive, anti-social, and thrill-seeking (high psychopathy) do not use swear words frequently if they also have low tendency towards manipulativeness (i.e., low Machiavellianism) and at the same time high tendency to grandiosity (i.e., high narcissism). Last but not the least, the association psychopathy-swear words' offensiveness was not a clear-cut. High levels of psychopathy were associated to low offensiveness *only* when both narcissism and Machiavellianism were high (MNP vs. MNp).

## STUDY 2: SOCIOSEXUALITY

### Method

#### Participants and procedure

Participants in Study 2 were also recruited from Amazon's Mechanical Turk and got paid $ 0.50 dollars for taking the survey (data from *Haddad et al., 2016*). Participants answered to measures of the Dark Triad, sociosexuality, demographic questions (e.g., gender, age) and two control questions (e.g., "In this question please answer Neither agree or disagree"). Nine participants responded incorrectly to the control question and were therefore eliminated from the final sample (i.e., 2.91%). This sample consisted of 309 US-residents, 104 women and 205 men ($N_2 = 309$, $M_{age} = 30.97 \pm 9.63$).

#### Measures

*Dark Traits*. As in Study 1, the Dark Triad Dirty Dozen (*Jonason & Webster, 2010*) was used to measure the three dark character traits: Machiavellianism (*Cronbach's $\alpha$ = .82*), narcissism (*Cronbach's $\alpha$ = .77*), and psychopathy (*Cronbach's $\alpha$ = .77*).

*Sociosexuality*. The Sociosexual Orientation Inventory Revised (*Penke & Asendorf, 2008*) consists of nine statements that measure the three sociosexual dimensions: behavior with a nine-point scale ranging from *0* to *20 or more* and items such as "With how many different partners have you had sexual intercourse on one and only one occasion?" (*Cronbach's $\alpha$ = .79*); attitude with a nine-point scale ranging from 1 (*=strongly disagree*) to 9 (*=strongly agree*) and items such as "Sex without love is OK" (*Cronbach's $\alpha$ = .86*); and desire with a nine-point scale ranging from 1 (*=never*) to 9 (*=at least once a day*) and items such as "In everyday life, how often do you have spontaneous fantasies about having sex with someone you have just met?" (*Cronbach's $\alpha$ = .87*). A global sociosexual orientation composite (i.e., the sum of all three dimensions) was also calculated (*Cronbach's $\alpha$ = .87*).

#### Statistical procedure

The same procedure as in Study 1 was followed to create all the eight possible combinations of high and low dark trait *percentile* scores or Dark Triad profiles: MNP "maleficent" ($n_2$

= 67, 21.7%), MNp "manipulative narcissistic" ($n_2$ = 31, 10.0%), MnP "anti-social" ($n_2$ = 34, 11.0%), Mnp "Machiavellian" ($n_2$ = 23, 7.4%), mNP "psychopathic narcissistic" ($n_2$ = 15, 4.9%), mNp "narcissistic" ($n_2$ = 45, 14.6%), mnP "psychopathic" ($n_2$ = 24, 7.8%), and mnp "benevolent" ($n_2$ = 70, 22.7%).

## Results and discussion

Paired *t-tests* were used to investigate the differences in sociosexual orientation (behavior, attitude, desire, and global sociosexual orientation) between individuals with malevolent character profiles who differed in one of the dark character traits but were similar in the other two. As in Study 1, correlation analyses were also conducted in order to investigate the added value of the profile analyses in relation to linear correlations between the dark traits and sociosexuality.

The correlation analyses indicated that Machiavellianism was positively associated to all components of sociosexuality (Behavior: $r = .26$, $p < .001$; Attitude: $r = .28$, $p < .001$; Desire: $r = .31$, $p < .001$) and to the global sociosexuality composite ($r = .35$, $p < .001$). However, the dark profiles comparison showed that this was consistent *only* when individuals with a Machiavellian profile (Mnp) were compared to individuals with a benevolent profile (mnp). That is, high levels of Machiavellianism were associated to a tendency to frequently having encounters of uncommitted sex (i.e., high sociosexual behavior), promiscuous attitude to uncommitted sex (i.e., high sociosexual attitude), heightened sexual interest (i.e., high sociosexual desire), and high levels of global sociosexual orientation, *only* when both narcissism and psychopathy were low at the same time (Mnp vs. mnp). What is even more, this association was relatively more accentuated in the Dark Cube analyses where Cohen's *d* varied between 0.45 to 0.64. Hence, suggesting that the correlation analyses are correct for *only* those with a Machiavellian profile, which was 7.4% of the population in this study. Additionally, high levels of Machiavellianism were also associated to high levels of global sociosexual orientation when both narcissism and psychopathy were high (MNP vs. mNP). See Table 2.

Furthermore, the correlation analyses indicated that narcissism was positively associated to all components of sociosexuality (Behavior: $r = .17$, $p < .001$; Attitude: $r = .21$, $p < .001$; Desire: $r = .27$, $p < .001$) and to the global sociosexuality composite ($r = .27$, $p < .001$). However, the Dark Cube analyses indicated that high levels of narcissism were associated to frequently having encounters of un-committed sex (i.e., high sociosexual behavior), *only* when the other two dark character traits were high (MNP vs. MnP). Additionally, high levels of narcissism were associated to a promiscuous attitude to uncommitted sex (i.e., high sociosexual attitude), heightened sexual interest (i.e., high sociosexual desire), and high levels of global sociosexual orientation when both Machiavellianism and psychopathy were low (mNp vs. mnp). In other words, an individual with a narcissistic profile, approves and desires promiscuous sex encounters without emotional bonding, but actually does it *only* when she/he is also high in both Machiavellianism and psychopathy. That being said, the profile analyses showed that the linear correlations might *only* apply to individuals with a narcissistic profile, which was 14.6% of the population in this study.
**Table 2** **Results from the *t-tests* for each Dark Triad character trait for sociosexuality.** The black numbers indicate significant results.

| Dark Trait | Dark Profile | Behavior | | | Attitude | | | Desire | | | Global Sociosexual Orientation | | |
|---|---|---|---|---|---|---|---|---|---|---|---|---|---|
| | | *t* | *p* | Cohen's *d* | *t* | *p* | Cohen's *d* | *t* | *p* | Cohen's *d* | *t* | *p* | Cohen's *d* |
| | MNP vs. mNP | 1.93 | .06 | 0.43 | 1.74 | .08 | 0.39 | 1.32 | .19 | 0.29 | 2.17 | < .05 | 0.48 |
| | MNp vs. mNp | 0.84 | .41 | 0.19 | 0.57 | .57 | 0.13 | 0.14 | .89 | 0.03 | 0.64 | .52 | 0.15 |
| Machiavellianism | MnP vs. mnP | 0.87 | .39 | 0.23 | 0.88 | .38 | 0.24 | 1.50 | .14 | 0.40 | 1.39 | .17 | 0.37 |
| | Mnp vs. mnp | 3.03 | < .01 | 0.64 | 2.15 | < .05 | 0.45 | 2.23 | < .05 | 0.47 | 3.04 | <. 01 | 0.64 |
| | MNP vs. MnP | 2.31 | < .05 | 0.46 | 1.36 | .18 | 0.27 | 0.71 | .48 | 0.14 | 1.86 | .07 | 0.37 |
| | MNp vs. Mnp | −1.02 | .31 | −0.28 | −0.05 | .96 | −0.01 | 0.59 | .56 | 0.16 | −0.16 | .56 | −0.04 |
| Narcissism | mNP vs. mnP | 0.29 | .77 | 0.10 | 0.08 | .94 | 0.03 | 0.61 | .55 | 0.20 | 0.26 | .70 | 0.13 |
| | mNp vs. mnp | 0.83 | .41 | 0.16 | 2.00 | < .05 | 0.38 | 3.51 | < .001 | 0.66 | 2.75 | <. 01 | 0.52 |
| | MNP vs. MNp | 1.88 | .06 | 0.38 | 2.35 | < .05 | 0.48 | 1.72 | .09 | 0.35 | 2.63 | < .05 | 0.54 |
| | MnP vs. Mnp | −1.42 | .16 | −0.38 | 0.76 | .45 | 0.20 | 1.30 | .20 | 0.35 | 0.48 | .63 | 0.13 |
| Psychopathy | mNP vs. mNp | 0.16 | .87 | 0.04 | 0.45 | .64 | 0.12 | 0.09 | .93 | 0.02 | 0.31 | .76 | 0.08 |
| | mnP vs. mnp | 0.49 | .62 | 0.10 | 2.07 | < .05 | 0.43 | 2.07 | < .05 | 0.44 | 2.11 | < .05 | 0.44 |

**Notes.**

M/m, high/low Machiavellianism; N/n, high/low narcissism; P/p, high/low; MNP, maleficent; MNp, manipulative narcissistic; MnP, anti-social; Mnp, Machiavellian; mNP, psychopathic narcissistic; mNp, narcissistic; mnP, psychopathic; mnp, benevolent.

Finally, the correlation analyses indicated that psychopathy was positively associated to all components of sociosexuality (Behavior: $r = .14$, $p < .001$; Attitude: $r = .22$, $p < .001$; Desire: $r = .23$, $p < .001$) and to the global sociosexuality composite ($r = .25$, $p < .001$). However, the Dark Cube analyses indicated that high levels of psychopathy were associated to a promiscuous attitude to uncommitted sex (i.e., high sociosexual attitude), heightened sexual interest (i.e., high sociosexual desire), and high levels of global sociosexual orientation *only* when both Machiavellianism and narcissism were low (mnP vs. mnp). In addition, both sociosexual attitude and global sociosexual orientation were positively associated to psychopathy *only* when Machiavellianism and narcissism were low (MNP vs. MNp). These complex interactions also discerned that frequently having encounters of un-committed sex (i.e., high sociosexual behavior) was not related to psychopathy. In other words, that type of behavior was *only* associated to high levels of psychopathy, when the other two traits varied and psychopathy was constant.

## GENERAL DISCUSSION

The aims in this set of studies were to investigate the relationship between malevolent character traits and "dirty" behavior (i.e., swearing and sociosexual orientation) using non-linear methods (i.e., comparisons between individuals who differed in one malevolent character trait but were alike in the other). As shown by the discrepancies between the results from the correlation analyses and the results from the paired *t-test*, the Dark Cube approach considers the complexity of processes within the person (cf. *Cloninger & Zohar, 2011*). At a general level, the results of this nonlinear approach suggested that the frequent usage, not level of offensiveness, of swear words was associated to Machiavellianism and narcissism. In other words, individuals with high levels in these traits might swear and are verbally offensive often because they do not see swearing as offensive (cf. with the attitude-behavior-cognition-hypothesis of taboo words; *Rosenberg, Sikström & Garcia, 2017*). Moreover, a promiscuous sexual attitude and desire were related to each dark trait *only* when the other two were low. Additionally, promiscuous sociosexual behavior was not associated to these malevolent character traits. That is, individuals high in the dark traits are willing to and have the desire to engage in sexual relations without closeness, commitment, and other indicators of emotional bonding. However, they do not report high levels of previous sexual experience, relationships, and infidelity. Hence, they approve and desire for it, but they are not actually doing it.

Some important limitations are the fact that the present study was cross-sectional and that the data were self-reported and therefore subject to personal perceptual bias. Replication and longitudinal studies should therefore be the next step. In addition, future studies should be conducted by controlling for demographics, such as, education, age, and gender. That being said, recent studies show that individuals who score high in self-reported dark traits display a congruent identity to their self-reported scores. For example, individuals high in Machiavellianism describe themselves as sarcastic, those high in narcissism describe themselves as extroverted and leaders, while those high in psychopathy describe themselves as mean (e.g., *Garcia et al., 2020*). In addition, individuals' narratives have also been found

as predictive of personality traits (e.g., *Garcia & Sikström, 2014*; *Garcia, Kjell & Sikström, 2013*; *Garcia, Kjell & Sikström, 2014*; *Garcia, Kjell & Sikström, 2020*). Thus, self-reported dark character traits are probably good measures that are predictive of actual malevolent and "dirty" behavior (see also *Moradi et al., 2015*).

Nevertheless, the most important limitation is actually the measure we used here to operationalize the dark triad (i.e., the Dark Triad Dirty Dozen). First of all, we opted to use a 7-point Likert scale (cf. *Jonason & Luévano, 2013*), but other studies have used a 5-point Likert scale (e.g., *Jonason, Li & Czarna, 2013*; *Jonason, Slomski & Partyka, 2012*) or even a 9-point Likert scale (e.g., *Jonason & Webster, 2010*). This variation makes it difficult to compare samples, thus, our findings need to be replicated using more reliable measures of the Dark Triad (see also *Persson, 2019*). Secondly, the validity of the Dark Triad Dirty Dozen has been criticized (e.g., *Lee et al., 2013*; *Miller & Lynam, 2012*; *Paulhus & Jones, 2014*). Some studies have actually suggested that the Dark Triad Dirty Dozen actually measures a Dark Dyad: an anti-social trait (an amalgamation of Machiavellianism and psychopathy) and narcissism (e.g., *Garcia & Rosenberg, 2016*; *Kajonius et al., 2016*; *Persson et al., 2016*). In addition, besides the methodological issues with the Dark Triad Dirty Dozen, conceptually, human personality and specifically character involves how we view our-selves, our relationship with others and society, and our existence as a whole (*Cloninger, 2004*). This ternary awareness of the self (i.e., the self, others and something greater that the self) is a whole system unit that is biopsychosocial in nature (*Cloninger, 2004*). In this context, while narcissism may correspond to character in relation to the self and the antisocial amalgamated Machiavellianism-psychopathy trait may correspond to character in relation to others, the Dark Triad seems to lack a character that corresponds to spirituality or the view of the self in relation to something bigger that the self (*Garcia & Rosenberg, 2016*). If it is so, the traits included in the Dark Cube as a model of dark and malevolent character profiles need to be reconsidered.

Finally, it is plausible to argue that calling swearing and high scores in sociosexuality for "dirty" behaviors is a bit too much. That being said, swearing is commonly known as "dirty" language (*Jay, 2000*). Even the measure used here for the dark traits is known as the Dark Triad "Dirty" Dozen, that is referring to a "quick and dirty" measure. Nonetheless, promiscuous sexual behavior or high scores in sociosexuality are not necessary seen as "dirty" in all cultures and contexts (*Davis & Whitten, 1987*). Related to this, some might argue that the dark traits have been associated with some of humanities' greatest vices, but also greatest virtues. Thus, suggesting that whether the traits are dark or malevolent might be in the eye of the beholder or depends on the situation. For instance, some researchers have even depicted a romantic picture of an agentic "James Bond" character based on the association between high scores in the Dark Triad traits and being extraverted, open, emotionally stable, having high self-esteem, and with a more individualistic and competitive approach to others (e.g., *Jonason, Li & Teicher, 2010*). However, not only have these studies low replicability and sometimes even contradictory findings (see for example *Garcia, Rapp Ricciardi & Ambjörnsson, 2016*), but many of these studies do not take into consideration or discuss the fact that high agency without high communion (i.e., cooperation, empathy, social tolerance, and helpfulness) and spirituality (i.e., spiritual

acceptance and meaning beyond the self) does not lead to a virtuous life (*Cloninger, 2004*). Indeed, features such as hope, empathy, and respect for one's self and others emerges from a self-transcendent outlook on life with a sense of participation in the boundless unity of all things or inseparable connectedness with nature and other people (*Cloninger, 2004*; *Garcia et al., 2019*). In other words, even if a person who is high in dark traits is agentic, without compassion, she/he will always be self-serving and egocentric. Which in other words will lead to manipulation and other type of behavior that here is depicted as dark or malevolent.

## CONCLUDING REMARKS

The present investigation gives a nonlinear approach to the study of the Dark Triad traits and their "dirty" behavior, in this case swearing and high levels of sociosexuality. The use of person-centered and non-linear methods, such as the Dark Character Cube, are helpful in the advancement of a coherent theory of a biopsychosocial model of dark character. Human character is, after all, a complex dynamic adaptive system (*Cloninger, 2004*). As such, malevolent character should express the characteristics of multi-finality and equifinality. Others, however, have pointed out that darkness is just the absence of light, and that we probably need to investigate the lack or underdevelopment of light character traits in order to understand what makes individuals live a virtuous or a vicious life (*Garcia & Rosenberg, 2016*; cf. *Cloninger, 2004*).

## ACKNOWLEDGEMENTS

I would like to thank Fernando R. Gonzáles Moraga at the Department of Psychiatry, Lund University, for his comments on the first drafts of this paper and for his help calculating Cohen's d in both studies.

### Funding
The author received no funding for this work.

### Competing Interests
Danilo Garcia declares that he has no competing interests. Danilo Garcia is the Head of Research of the Blekinge Center of Competence, which is the Region Blekinge's research and development unit. The Center works on innovations in public health and practice through interdisciplinary scientific research, community projects, and the dissemination of knowledge in order to increase the quality of life of the habitants of the county of Blekinge, Sweden. Danilo Garcia is also associate professor at Linköping University and the University of Gothenburg, and the chairman of the Network for Well-Being. This is a nonprofit international network of junior and senior researchers that support students and researchers who are interested in the Science of Well-Being.

## Author Contributions

- Danilo Garcia conceived and designed the experiments, performed the experiments, analyzed the data, prepared figures and/or tables, authored or reviewed drafts of the paper, and approved the final draft.

## Human Ethics

The following information was supplied relating to ethical approvals (i.e., approving body and any reference numbers):

This work was not subject to ethical review according to Swedish law.

## Data Availability

Raw data is available in the Supplemental Files.

## Supplemental Information

Supplemental information for this article can be found online at http://dx.doi.org/10.7717/peerj.9620#supplemental-information.

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
