# Peer review of "How “dirty” is the Dark Triad? Dark character profiles, swearing, and sociosexuality"

_PeerJ, doi:10.7717/peerj.9620_

## Round 0.1 · original submission · Major Revisions

Thank you for your submission, both reviewers have raised a number of concerns that should be addressed.

Reviewer 1 ·

Basic reporting

The article does not provide sufficient introduction and background to demonstrate how the work fits into the broader field of knowledge. In particular, it would be helpful to explicitly locate this work theoretically within the context of evolutionary psychology approaches to sexuality (with all the assumptions that this contains). Without explaining this to the reader, some of the text is difficult to follow – especially for a reader unfamiliar with this approach. For example, in the following text it is not clear what is meant by an ‘exploitative short-term mating strategy’ or ‘mate-value’:
75 dark traits are expected to use an exploitative short-term mating strategy that is associated with 76 strategies and tactics to avoid entangling commitments in mate-value
In addition, key terms are used interchangeably and without explanation within the background review of the literature. For example:
• In what ways does promiscuous behaviour represent an ‘outlook of separation’ (line 61).
• Why are these character traits labelled as ‘dark’ or ‘malevolent’ when they can also be conceptualised as include positive or socially useful qualities?
• The malevolent character is also labelled as is labelled as ‘immature’ without explanation (line 59).
There are a set of value judgements about sexuality in operation throughout this paper which are not explored or articulated. For example, swearing and promiscuous sexual behaviour are both identified as ‘dirty’ behaviour without comment or explanation (line 113). However, these are not self-evidently ‘dirty’.
The paper would also benefit from a more clearly articulated rationale. Why is it important for us to know whether people with different combinations of personality/character traits engage in more swearing and/or view swearing as less offensive? Why is it important for us to know whether people with different combinations of personality/character traits are more willing to engage in sexual behaviours without commitment or closeness?

Experimental design

It is not clear whether this paper fits the remit of PeerJ to publish articles in Biological Sciences, Environmental Sciences, Medical Sciences, and Health Sciences. The authors do not explicitly connect their work to any of these areas.

It would be useful to provide some information about the psychometric properties of the different measures used.

The study has not been subject to ethical review as the author state that this is not required within Sweden for this type of study.

Validity of the findings

The discussion of the results would have benefited from interpretation in the light of the original research questions and the underlying theory.

Reviewer 2 ·

Basic reporting

The references in the text is not in accordance with references given in the reference section. There are very many discrepancies. Sometimes the reference is lacking in the text and sometimes it is lacking in the reference section. My guess is that something went wrong when writing the reference section that is easy to correct.

The reference to Garcia & González says 2017 in the text but 2016a in the reference section. It is difficult to find the reference in Time and Society since there is no issue. The reference was not found in a journal search between 2015 and 2017.

Experimental design

The purpose and objectives of the two studies are described, but it is difficult to discern a clear research question.

If the research question is the same as the purpose, it should be clarified.

Validity of the findings

The general discussion is well written and very interesting.

One suggestion is to describe the result of this study more clearly. The discussions under headings "result and discussion" for each studie could be more extensive.

Additional comments

Thank you for an interesting and well written study on the Dark Cube and the relation between the dark traits and promiscuous behavior.

The proposed replications may further deepen the knowledge of the dark triad. A comparison with a dark dyad instead of triad is also an interesting idea.

---

## Round 0.2 · accepted · Accept

Thank you for your resubmitted paper. I am delighted to inform you that this can now be accepted for publication.

Reviewer 2 ·

Basic reporting

Comments on reference errors are corrected

Experimental design

The questions on research question and purpose are answered. .

Validity of the findings

The results are clearified and expanded and so is the discussion

Additional comments

No comment